# Characterizing the Heart Rate Response to the 4 × 4 Interval Exercise Protocol

**DOI:** 10.3390/ijerph17145103

**Published:** 2020-07-15

**Authors:** Justin J. Acala, Devyn Roche-Willis, Todd A. Astorino

**Affiliations:** Department of Kinesiology, California State University, San Marcos, CA 92096-0001, USA; acala001@cougars.csusm.edu (J.J.A.); roche016@cougars.csusm.edu (D.R.-W.)

**Keywords:** high intensity interval training, heart rate, exercise prescription, intermittent exercise

## Abstract

High intensity interval training is frequently implemented using the 4 × 4 protocol where four 4-min bouts are performed at heart rate (HR) between 85 and 95% HR max. This study identified the HR and power output response to the 4 × 4 protocol in 39 active men and women (age and VO_2_ max = 26.0 ± 6.1 years and 37.0 ± 5.4 mL/kg/min). Initially, participants completed incremental cycling to assess VO_2_ max, HR max, and peak power output (PPO). They subsequently completed the 4 × 4 protocol, during which HR and power output were monitored. Data showed that 12.9 ± 0.4 min of 16 min were spent between 85 and 95% HR max, with time spent significantly lower in interval 1 (2.7 ± 0.6 min) versus intervals 2–4 (3.4 ± 0.4 min, 3.4 ± 0.3 min, and 3.5 ± 0.3 min, *d* = 2.4–2.7). Power output was highest in interval 1 (75% PPO) and significantly declined in intervals 2–4 (63 to 54% PPO, *d* = 0.7–1.0). To enhance time spent between 85 and 95% HR max for persons with higher fitness, we recommend immediate allocation of supramaximal intensities in interval one.

## 1. Introduction

High intensity interval training (HIIT) is characterized by repeated efforts (1–4 min in duration) at near-maximal to maximal intensities interspersed with recovery [1]. Compared to moderate intensity continuous training (MICT), completion of HIIT induces superior increases in maximal oxygen uptake (VO_2_ max) in healthy adults [2], as well as persons with chronic disease [3], which benefits health, considering the strong relationship between VO_2_ max and mortality [4]. For example, in inactive men, Nybo et al. [5] demonstrated greater increases in VO_2_ max in response to 12-weeks of HIIT than that acquired with MICT. Results in obese young women also exhibit significantly greater increases in VO_2_ max and reductions in fat mass with sprint interval training compared to MICT of equal energy expenditure [6].

Despite the documented efficacy of HIIT, there is no single protocol that is universally utilized in a research or applied setting. For example, HIIT workloads as low as 60% of peak power output (% PPO) were utilized in sedentary populations to elicit increases in oxidative capacity [7] and VO_2_ max and fat oxidation [8]. In active adults, intensities above 80 [9] all the way up to 100% PPO [10] have been applied to enhance VO_2_ max and mitochondrial biogenesis. Number of bouts completed per session also varies from 4 to 6 bouts in persons with diabetes [11] to as many as 10 bouts in habitually active individuals [9,10]. This versatility of HIIT provides the clinician or scientist with an infinite number of permutations to employ. Some data show that dissimilar structures of HIIT may elicit different increases in VO_2_ max [12], although this finding is not universal [9].

One widely-used HIIT paradigm is the 4 × 4 regimen which requires four 4 min bouts at intensities eliciting 85–95% maximal heart rate (HRmax) interspersed with 3 min active recovery at 70% HR max. In moderately active individuals, Helgerud et al. [13] reported that an 8-week regimen of 4 × 4 running at 90–95% HR max led to superior increases in VO_2_ max versus MICT or training at lactate threshold. In obese participants, Baekkerud et al. [14] showed that 6-weeks of treadmill-based 4 × 4 training at 85–95% HR max significantly increased VO_2_ max by 10%, which was higher than observed in response to MICT or lower-volume HIIT. In patients with coronary artery disease, 10-weeks of 4 × 4 at 85–95% HR max elicited superior increases in VO_2_ max versus work-matched MICT [15]. Overall, this particular protocol is extremely effective to augment increases in cardiorespiratory fitness in various populations.

One requirement of this 4 × 4 protocol is that work rate must be continuously modified during each interval to achieve the target HR between 85 and 95% HR max. This differs from other widely-used interval protocols, such as the 10 × 1 min regimen [7,8], which is implemented based on a specific % PPO, as well as various sprint interval training regimens, which have a fixed load based on body mass [9]. Consequently, there is a certain amount of uncertainty in allocating power output, especially during the first 4-min interval, to achieve the desired intensity. In addition, there are concerns that increasing the work rate excessively at any point during exercise may elicit premature fatigue in the participant and inability to complete the session. Unfortunately, standard instructions regarding how to modify workload during the 4 × 4 are relatively unknown, with exception of one study performed in a small sample of active young men [16]. Data showed that power output is reduced during the session to keep participants’ HR between 90 and 95% HR max, and average work rate during the fourth interval (60% PPO) was significantly lower than in the first interval (75% PPO). However, they did not report how much time was spent in the desired HR range during each interval, which may be critical to augment resultant training adaptations [17]. Moreover, they used male participants, so data cannot be applied to women who may have different tolerance to intense exercise. Consequently, it is unknown how to adequately implement the 4 × 4 protocol in a more diverse population, and more study is merited to better understand this topic which applies to clinicians and sport scientists.

This study characterized the HR and power output response to a session of 4 × 4 cycling in a heterogeneous sample to attempt to develop another protocol applicable to healthy, young active adults. Creation of this protocol may reduce the guesswork of allocating intensities based on HR by identifying specific power outputs for clinicians to employ to elicit HR between 85 and 95% HR max. We also measured time spent between 85 and 95% HR max during the session and identified predictors of this duration to monitor if various measures of submaximal (ventilatory threshold) and maximal exercise capacity (VO_2_ max and PPO) are related to this outcome. It is hypothesized that, to maintain HR in the desired range, power output will be highest in interval 1 and decrease in subsequent intervals.

## 2. Materials and Methods

### 2.1. Design

VO_2_ max, HR max, and PPO were measured in the first session. The following session, participants performed the 4 × 4 protocol. Power output, heart rate, and blood lactate concentration were measured during the session. Before both sessions, participants were instructed to be well-rested and hydrated, fast for 2 h, and to refrain from intense exercise the day before. Sessions were held at the same time of day within participants and were separated by a minimum of 48 h and maximum of 7 days.

### 2.2. Participants

Thirty-nine healthy adults participated in this study. Participant demographics are listed in Table 1. Inclusion criteria consisted of completing more than 150 min of physical activity per week in the last year, lack of knee pain, and not being obese. These adults participated in resistance training, aerobic exercise, non-competitive sport, and/or rowing, yet none was athletic or training for a particular sport. All subjects provided informed written consent before starting this study (protocol 2019-121), which was approved by the University Institutional Review Board. The study was carried out in accordance with rules of the Declaration of Helsinki.

### 2.3. Baseline Session

All experiments were conducted in the Human Performance Laboratory with temperature and relative humidity equal to 20–22 °C and 30–50%, respectively. Height (in centimeters) and body mass (in kilograms) were initially measured to determine body mass index. The seat and handlebar height of an electronically braked cycle ergometer (Velotron Dynafit Pro; Racermate, Seattle, WA, USA) was recorded for each participant during the first session and applied subsequently.

Participants were connected to a calibrated metabolic cart (Parvo Medics TrueOne, Sandy, UT, USA) that measured pulmonary gas exchange data every 15 s. The reliability of this system, according to the manufacturer, is 0.1% for oxygen and carbon dioxide, respectively. A monitor (Polar Electro, Kempele, Finland) was placed around the trunk to measure HR. Power output began at 40–60 W for 2 min followed by 20–30 W/min increments in work rate based on their body size and fitness level until onset of volitional exhaustion (pedal cadence < 50 rev/min). VO_2_ max was identified as the mean of the two highest values in the last 30 s of exercise and was confirmed using standard criteria [18]. HR max and PPO were identified as the values coincident with exhaustion. Ventilatory threshold was determined using the ventilatory equivalents method [19] and was identified by two independent investigators.

### 2.4. 4 × 4 Protocol

Participants warmed up on the cycle ergometer for four minutes at 25% PPO. During exercise, they were allowed to maintain a self-selected cadence which was above 50 rev/min. Participants performed the 4 × 4 protocol consisting of four 4-min intervals at 85–95% HR max interspersed by 3 min recovery periods at 25% PPO. Power output was continually adjusted in steps of ±5% PPO to elicit and maintain 85–95% HR max for as much of each 4-min interval as possible. At the onset of each interval, power output was immediately increased to 85–100% PPO to elicit HR > 85% HR max. Once heart rate attained 95% HR max, power output was gradually reduced to slow rise in HR, promote exercise tolerance, and not induce premature fatigue. At any point during any interval that HR declined by ≥2 b/min, work rate was increased by 5% PPO. Heart rate and power output were continuously monitored and recorded every 15 s throughout the session, excluding the recovery periods.

### 2.5. Assessment of Blood Lactate Concentration (BLa)

Blood samples were obtained from the fingertip using a lancet (Owen Mumford, Inc., Marietta, GA, USA) and portable monitor (Lactate Plus; Sports Research Group, New Rochelle, NY, USA) at rest, immediately after bout 2, and 3 min after exercise to assess changes in BLa. The hand was cleaned with a damp towel, dried, and the first blood sample was wiped away.

### 2.6. Statistical Analysis

Data were analyzed using SPSS Version 24.0 (IBM, Armonk, NY, USA) and are reported as mean ± SD. The Shapiro-Wilks test was used to assess normality. We used a one-way repeated measures ANOVA to determine significant differences in HR, power output, time spent > 85% HR max, and BLa, and Tukey’s post hoc test was used to identify differences between means if a significant F ratio occurred. Cohen’s d was used as an estimate of effect size, and 95% confidence intervals (95% CI) were used as appropriate. Pearson product moment correlation coefficient was used to examine associations between variables. To elucidate the day-to-day variability in our measures and best characterize the HR and PO response to 4 × 4 in an attempt to develop a protocol that can be used in a similar population, three men of similar age, body mass, and VO_2_ max as our sample performed the 4 × 4 protocol at the same time of day on three separate days. Other studies on this topic [16,20] did not present reliability data, and we believed that testing only 3 individuals would provide enough data to exhibit the stability in these outcomes across days. Total duration between 85 and 95% HR max and mean HR during intervals 1–4 were not different (*p* > 0.05), highly related (ICC = 0.75–0.95), and yielded minimal differences in time (0.5 min for total time and 0.1 min in each interval) and HR (<2 b/min) across trials. Statistical significance was equal to *p* < 0.05.

## 3. Results

### 3.1. Change in HR during the 4 × 4 Protocol

Compared to the warm-up, HR significantly increased (*p* < 0.001) during exercise and peaked at 172 ± 9 b/min during interval 4, representing 93% HR max. Figure 1 shows participants’ average HR response to each interval with mean HR equal to 160.3 ± 11.0, 166.5 ± 11.0, 167.0 ± 10.0, and 169.0 ± 10.0 b/min, respectively, during each interval. HR gradually increased during each interval, and the HR value upon initiation of each interval steadily increased during the session. Post hoc analyses showed that there was a significantly higher average HR in response to the last three intervals versus interval 1 (*d* = 1.4).

Time spent between 85 and 95% HR max increased during the session (*p* < 0.001), and post hoc analyses showed that time in intervals 2 through 4 was significantly higher than in interval 1 (*d* = 2.4–2.7) (Table 2). Women exhibited a significantly higher (*p* = 0.003) duration spent between 85 and 95% HR max versus men (13.5 ± 1.1 min vs. 12.4 ± 1.2 min, *d* = 1.1).

### 3.2. Change in Power Output during the 4 × 4 Protocol

Power output significantly declined (*p* < 0.001) during the session. Figure 2 shows the average power output in response to each interval and the overall response for all participants. Power output was increased in the first 1–2 min of each interval, after which it was reduced to not surpass 95% HR max. Post hoc analyses showed that interval 1 (195.0 ± 58.0 W) exhibited a significantly higher power output compared to subsequent intervals (161.2 ± 48.0, 154.3 ± 44.0, and 149.8 ± 48.0 W, respectively) (*d* = 0.7–1.0).

### 3.3. Change in Blood Lactate Concentration

There was a significant increase in BLa (*p* < 0.001) from rest (1.4 ± 0.5 mM) to mid-bout (9.3 ± 2.4 mM) and post-exercise (8.1 ± 2.2 mM). Post hoc analyses showed that the resting value was lower than BLa acquired during exercise (*d* = 4.7. and 5.6, respectively) and that the two exercise values were significantly different from each other (*d* = 0.90).

### 3.4. Correlation Analyses

Results showed no relationship between total time spent between 85 and 95% HR max and VT (*r* = 0.05, *p* = 0.78) or total change in BLa (*r* = −0.24, *p* = 0.14). However, there was a significant relationship between time spent between 85 and 95% HR max and PPO (*r* = −0.47, *p* = 0.002), HR recorded 15 s into interval 1 (*r* = 0.58, *p* < 0.001), and absolute VO_2_ max (*r* = −0.47, *p* = 0.003).

## 4. Discussion

This study used a large, active sample diverse in gender and fitness level to characterize the HR and power output response to the 4 × 4 protocol. Despite the widespread use of the 4 × 4 protocol to enhance health status and cardiorespiratory fitness in various populations [14,15,16], there is no clear method to implement this in daily practice since previous studies rarely describe how intensity is manipulated during exercise to achieve the desired HR range. Our data showed that time spent between 85 and 95% HR max ranges from 66 to 100% across participants and increased from the first to the last interval. In addition, there was a significant reduction in power output from interval 1 to 4, and the fourth interval elicited slightly greater than 50% PPO.

Little research has elucidated the exact modification in work rate needed during the 4 × 4 to achieve the desired intensities. For example, Helgerud et al. [13] required active men to complete the 4 × 4 on a treadmill at 90–95% HR max. Their methods stated that “the subjects were instructed to reach the target intensity in about 1–1.5 min and then stay there by adjusting the speed of the treadmill”. However, this does not explain how much or how rapidly intensity was modified to achieve the desired HR nor does it indicate how intensity was changed during each interval to ensure that speeds eliciting HR > 95% HR max were not attained. Consequently, this rather incomplete text does not allow the practitioner to repeat this methodology, and, in turn, he or she is left guessing as to the specific intensity to utilize in interval 1 to achieve the target intensity, and, in subsequent bouts, to maintain or not surpass it.

Our results show that approximately 80% of the 4 × 4 protocol is spent in the desired intensity range, with time spent between 85 and 95% HR max significantly increasing from the first interval to all subsequent intervals. Our overall time spent in this zone in greater than that demonstrated in active men completing the 4 × 4 protocol on a cycle ergometer. Falz et al. [20] reported that only 68 ± 8% of the session was spent between 85 and 95% HR max, with time in this zone equal to 46 ± 22% for interval 1 and 69 ± 12 for interval 2. Their warmup was completed at 50% PPO, which should increase HR more so than that used in the current study (25% PPO), so we have no explanation for this discrepancy. Much of the challenge at the onset of the 4 × 4 is the large disparity in HR values between the warm-up (work rate = 65 W eliciting a heart rate of approximately 110 b/min in the present study) and interval 1, in which HR had to attain a minimum value equal to 156 b/min across our participants. There is also an inherent physiological lag associated with this measure [17]. When increasing power output to achieve this target HR, we elected to be somewhat conservative and did not require our untrained participants to sprint or cycle at supramaximal work rates, which we believed may induce premature fatigue and diminish tolerance to the subsequent three intervals. Therefore, it took participants approximately 1.3 min to attain 85% HR max, although this occurred in less time in participants with lower VO_2_ max whose warm-up HR was not substantially lower than that attendant with interval 1. For example, a woman with VO_2_ max, PPO, HR max, and HR warm-up equal to 31.6 mL/kg/min, 3.4 W/kg, 197 b/min, and 146 b/min spent 94% of the session in the desired HR zone, and her HR at the end of recovery 1 was only 10 b/min lower than the value at the end of the first interval (189 b/min). In contrast, a woman with VO_2_ max, PPO, HR max, and HR warm-up equal to 43.4 mL/kg/min, 4.3 W/kg, 188 b/min, and 120 b/min spent 77% of the session in the desired HR zone, which was attributed to a marked decline in HR (30–35 b/min) in recovery and gradual rise upon initiation of each interval. Although not measured in this study, this change in HR is likely regulated by the balance of the parasympathetic and sympathetic response. Compared to more fit adults [21], it is possible that less fit participants have greater sympathetic responses leading to more rapid increases in HR in response to exercise and lower reductions in HR during recovery, ultimately eliciting more time spent between 85 and 95% HR max. In addition, the onset of cardiovascular drift may also contribute to acceleration in HR in the latter part of the session, as this intensity is between the participant’s threshold and VO_2_ max [22].

Our results show that power output was substantially reduced from interval 1 (75% PPO) to interval 2 (63% PPO) and gradually decreased in subsequent intervals to 59 and 54% PPO, yet HR was maintained in the desired zone and in fact, continued to rise despite this decrement in power output. This result was also reported by Tucker et al. [16], who demonstrated an identical power output during interval 1 (75% PPO) and slightly higher power outputs during intervals 2–4 (68, 62, and 60% PPO). This discrepancy is explained by their slightly higher intensity (90–95% HR max) and their male subjects’ higher VO_2_ max and in turn, higher PPO that is attendant with a greater decline in HR in recovery that requires a larger increase in work rate. Falz et al. [20] reported a mean work rate equal to 68% PPO during completion of the 4 × 4 protocol, although they did not report values for each interval. We recommend that, for participants with average to above average VO_2_ max, as used in Tucker et al. [18], power outputs slightly higher than our values be used; whereas, in subjects with below average VO_2_ max, using similar or slightly lower work rates will elicit substantial duration between 85 and 95% HRmax and promote completion of the protocol.

Blood lactate concentration peaked after interval 2 and was equal to 9.3 mM, although values ranging from 4.8 to 12.5 mM were demonstrated across participants. These values parallel those previously reported in response to the 4 × 4 protocol equal to 8.5 and 12.1 mM [16,20] and emphasize substantial contribution of glycolysis towards adenosine triphosphate (ATP) supply during the session. Interestingly, BLa declined in the latter half of the session, which is likely related to the reduction in power output, as well as oxidation of lactate [23].

The present study has some limitations. First, data cannot be applied to older, inactive, or highly trained populations whose HR response and tolerance to interval exercise such as 4 × 4 differ. Consequently, we encourage scientists to initiate studies like ours to clearly describe patterns of work rate modification to achieve the target HR attendant with 4 × 4 in these select groups. In addition, it is likely that higher relative power outputs are needed to implement the 4 × 4 during treadmill running compared to cycle ergometry, due to the higher HR observed during cycling compared to running [24,25], which may be related to a greater sympathetic response [26]. We also did not measure gas exchange data during the 4 × 4 protocol, so we are unsure as to the specific fraction of VO_2_ max elicited by this session. However, it elicits 80% VO_2_ max in active men [16]. Nevertheless, this study is strengthened by recruitment of a large and rather heterogeneous sample, use of relatively small changes in power output, and measurement rather than prediction of HR max.

## 5. Conclusions

Overall, methods and data from this study provide practitioners with a feasible approach to implement the 4 × 4 protocol in men and women with average cardiorespiratory fitness. Despite a reduction in power output during the session, HR was between 85 and 95% HR max for more than 80% of the bout, and it is likely that less fit individuals will spend more time in this zone.

## Figures and Tables

**Figure 1 ijerph-17-05103-f001:**
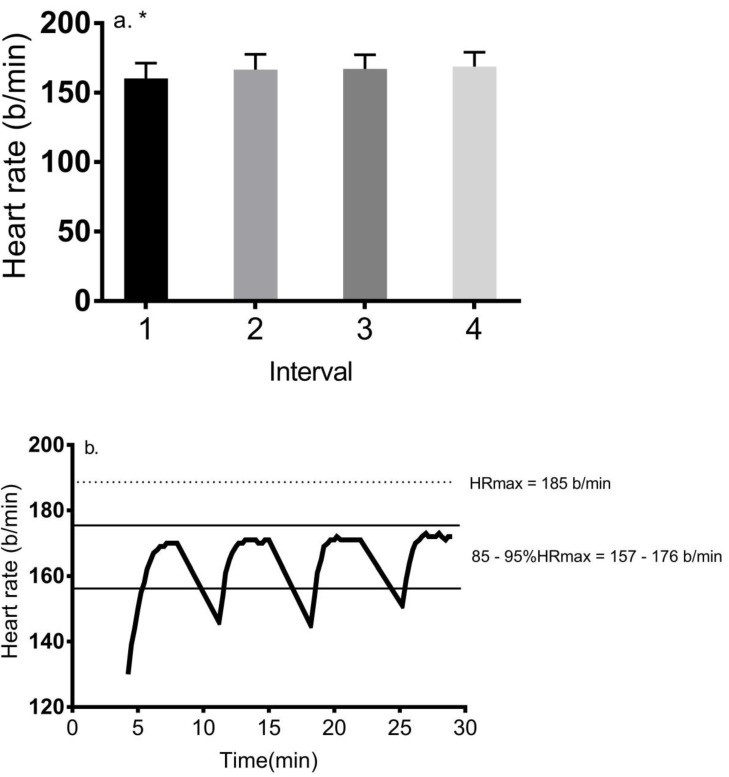
(**a**) Mean heart rate response to each of the four intervals during the 4 × 4 protocol (mean ± SD). * = *p* < 0.05 for interval 1 versus intervals 2–4; (**b**) overall HR response during the 4 × 4 protocol for all 39 participants. Note that minutes 4–8, 11–15, 18–22, and 25–29 represent the four intervals of the session. The 2 lines represent the HR zone from 157 to 176 b/min, representing 85–95% HR max.

**Figure 2 ijerph-17-05103-f002:**
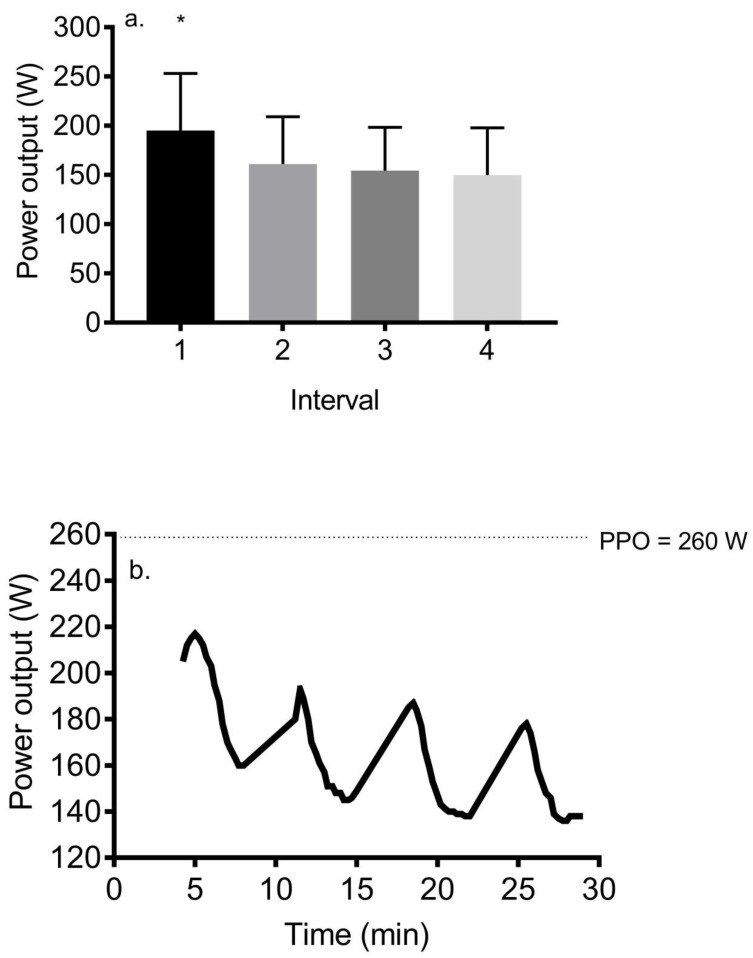
(**a**) Mean power output response to each of the four intervals during the 4 × 4 protocol (mean ± SD). * = *p* < 0.05 versus intervals 2–4; (**b**) overall power output response during the 4 × 4 protocol for all 39 participants. Note that minutes 4–8, 11–15, 18–22, and 25–29 represent the four intervals of the session.

**Table 1 ijerph-17-05103-t001:** Physical characteristics of participants, including maximal gas exchange data (mean ± SD).

Variable	Total
Number (M/F)	20/19
Age (year)	26.0 ± 6.1
Mass (kg)	72.1 ± 10.8
BMI (kg/m^2^)	24.2 ± 2.9
VO_2_ max (mL/kg/min)	37.0 ± 5.4
VO_2_ max (L/min)	2.7 ±0.6
VCO_2_ max (L/min)	3.3 ± 0.6
RER max	1.30 ± 0.09
V_E_ max (L/min)	112.5 ± 24.5
HR max (b/min)	184.9 ± 8.6
PPO (W)	260.1 ± 41.5
Physical activity (h/week)	5.4 ± 2.4

M = male; F = female; BMI = body mass index; VO_2_ max = maximal oxygen uptake; VCO_2_ max = maximal carbon dioxide production; RER = respiratory exchange ratio; V_E_ = ventilation; HR = heart rate; PPO = peak power output.

**Table 2 ijerph-17-05103-t002:** Time spent between 85 and 95% HR max during the 4 × 4 regimen (mean ± SD).

Interval Number	Time Spent between 85 and 95% HR Max (min)	95% CI
1	2.7 ± 0.6 *	2.5–2.8
2	3.4 ± 0.4	3.2–3.6
3	3.4 ± 0.5	3.2–3.6
4	3.5 ± 0.3	3.2–3.6

HR = heart rate; CI = confidence interval; * = *p* < 0.05 versus all other intervals.

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
