# Peer review of "Characterizing the Heart Rate Response to the 4 × 4 Interval Exercise Protocol"

_ijerph, 2020, doi:10.3390/ijerph17145103_

Round 1

Reviewer 1 Report

Dear authors,

I congratulate the authors for writing the manuscript. This manuscript provides interesting results in the field of physical fitness.

I provide you with several suggestions to improve the quality of the manuscript.

Introduction: Add the study hypotheses at the end of the introduction section.

4x4 protocol: Are there researches that analyze the reliability of the instruments (polar electro, calibrated metabolic cart, etc.) used in this study? Are there previous researches that have used these instruments?

Discussion: Add the study purpose in the first paragraph.

Would you recommend the 4x4 protocol to train a specific sport?

Author Response

General comments:  We extend to the reviewers sincere appreciation for a thorough and objective review of our submission, which is of interest to the field considering the wealth of information documenting efficacy of interval training using this particular 4X4 protocol, yet little knowledge concerning how to actually implement it in a clinical or fitness setting.   In the point-by-point rebuttal below, we have attempted to respond to all concerns raised by the reviewers, with revised text denoted in italics.  In addition, we have made various changes to our manuscript in blue font.  Please note that some line numbers have changed due to these revisions.  Overall, we hope that these modifications are sufficient to vastly improve this paper to merit publication in your Journal.

Specific comments, Reviewer #1:  We appreciate this reviewer’s feedback praising our work and in addition, for his-her belief that these are interesting results towards the field of physical fitness.  Below, we have responded to each of his/her initial concerns regarding our submission.

Introduction: Per your comment, this hypothesis has been added to the end of the Introduction (page 2 line 71). Thank you and we hope that this change satisfies your concern.  It is hypothesized that to maintain HR in the desired range, power output will be highest in interval 1 and decrease in subsequent intervals.     

4X4 protocol:  You raise a very fair point about the reliability of the ParvoMedics metabolic cart as well as for the Polar HR monitor used in the current study. We believe that the reliability of the metabolic cart is not that relevant, especially considering that we did not acquire gas exchange data during the 4X4 session.  However, it is apparent that estimates of VO2 and VCO2 from this system have an accuracy of 0.1 %, which is exceptional.  As for the HR telemetry system by Polar, our current text page 4, line 136 states that repeated testing in 3 men provided reliability data of < 2 b/min for this device.  So, we are confident in the reliability of our HR assessment to identify small changes in HR in response to the 4X4 session. Thank you and we hope that this explanation is adequate.   

Discussion:  Per your comment, the text below was added to the start of the Discussion on page 7, line 175. Thank you and we hope that this change appeases your concern.  This study characterized the HR and power output response to the 4 X 4 protocol. 

Use of the 4X4 protocol for training:  This is a fair point and it is our belief that any sport requiring an adequate VO2max and lactate/ventilatory threshold would benefit from inclusion of this regimen into regular training.  Moreover, it forces the exerciser to maintain vigorous intensities for a relatively long time, which mirrors certain intermittent sports. Although we are unaware of specific empirical data in this area, we assert that athletes participating in hockey, basketball, wrestling/MMA, boxing, cycling, and potentially even soccer or American football would benefit. We hope that this response is satisfactory; thank you.

Reviewer 2 Report

Dear Authors,

After reading the manuscript, below you will find my suggestions:

Abstract

Remove subtitles (objectives, methods…)

Introduction

There is no sufficient evidence reported about HIIT, benefits, and comparison to other training methods. Please, expand.

You claimed that “there is no single protocol that is universally utilized in a research or applied setting”. I don’t feel this is necessarily a bad thing, as the percentage of peak power can be adapted to the population. In fact, you mentioned studies (again scarce evidence) of sedentary to active populations. You also acknowledged in the conclusion section, that your results cannot be applied to older, inactive, or highly trained populations or even to other types of instruments (treadmill running compared to cycle ergometry). Therefore, your proposed protocol does not solve the problem of standardized method, but to a very limited group in a spectrum of possibilities. I suspect that the reason behind the different thresholds in the literature account to be adapted to the population and instrument with which it is being measured. 

Your objective is somewhat too pretentious  “This study characterized the HR and power output response to a session of 4 X 4 cycling in a heterogeneous sample to attempt to develop a protocol applicable to healthy, young active adults.” I would say “to attempt to develop ANOTHER protocol” because your proposal might be seen as good as the other studies you mentioned.

Materials and methods

Table 1: What is the point of comparing male and female athletes?

In the statistical analysis, it is implied that 3 participants performed the protocol three times. Is that something belonging to the experiment? What did the other participants?

Results

Figure 1 shows the HR response. I can’t see the exact values of the four intervals but it seems very alike, so I don’t understand the significant differences between HR values. What are you comparing? 2 vs 4? Why only these intervals? Why you mentioned that HR significantly increased (p < 0.001) during exercise?

If your aim was to check the Time spent between 85 – 95 %HRmax, a table with these values would have been very interesting. In fact, I would change Fig 1a for a Table. Please, re-elaborate Figure 1b with more accuracy on the HR line (thick line with little detail) and better axis. Finally, consider the 85 – 95 %HRmax individually against the individual response and depict mean. Did you do it like this?

The same for the rest of the variables.

Discussion

During the discussion section, you emphasized that the work of others doesn’t allow repeating methodology and present your results as a solution. My suggestion here is to change the way your results are discussed. You should present them as another experiment, not as the solution. If you were to demonstrate your protocol to be the standardized 4x4 protocol, you should have done a randomized crossover trial with subjects performing at different 4x4 protocol designs to see which is the ultimate protocol so as to name it as standard.

Author Response

Specific comments, Reviewer #2:  We hope that the responses below and corresponding changes made to the paper satisfy your concerns regarding this paper; thank you.

Abstract: Our understanding is that the Journal Guidelines require these subheadings here, so we do not believe it is proper to remove them. Thank you.

Introduction: Your point regarding comparison of benefits accrued via HIIT versus other methods is fair, yet page 1 lines 25-28 denotes that results from recent systematic reviews suggest that the change in VO2max is greater in response to HIIT versus moderate intensity continuous training (MICT), which is the alternative to HIIT.  Moreover, lines 34-36 on this page state that adaptations accrued in response to various interval training protocols are often similar, yet not always.  We have published data from 3 longitudinal training studies, and for the most part, they show similar adaptations to HIIT in active (Astorino et al. 2017 MSSE) and inactive populations (Astorino et al. 2013 MSSE; Clark et al. 2019 EJAP) irrespective of intensity or volume of training which varied in each study.  We hope that this explanation appeases your fair concerns; thank you.

Our claim that “there is no single protocol that is universally utilized in a research or applied setting:” Yes, this is not a bad thing as long as those who choose to replicate these protocols in their own workplace or setting institute them properly. As our subsequent text states (page 2 lines 48-51), it is very easy to prescribe sprint interval training via repeated Wingate tests or in the case of HIIT, use some fraction of PPO; however, the 4X4 and other protocols relying on HR have tremendous uncertainty regarding what specific workload to assign to attain the target HR range.  Hence, we tried to elucidate this in the current study; we hope that this explanation appeases your fair concern; thank you.

Yes, we agree with you that our data can only be applied to active young men and women performing the 4X4 protocol on the cycle ergometer and not to any other population or exercise modality. Yet, we believe that our study is robust enough that a clinician could use our results to implement the 4X4 protocol if they worked with a similar population; thank you and we hope that this explanation appeases your concern.

Objective: Your point here is fair and we added this term to our Purpose statement. Thank you and we hope that this is satisfactory.  This study characterized the HR and power output response to a session of 4 X 4 cycling in a  heterogeneous sample to attempt to develop another protocol applicable to healthy, young active adults.

Materials and Methods, Table 1:  We include the physical characteristics of both sexes to be more candid to the reader, especially as we have a sample differing in sex.  We notice many articles doing the same, and since the other Reviewers had no issue with this, we do not believe it is merited to modify this. Thank you and we hope that this explanation satisfies your concern.

Statistical analysis:  This text refers to testing of 3 men of similar age and fitness level of the larger sample, who performed the 4X4 protocol three times to allow us to assess the reliability of our measures.  We believe that including this information strengthens our text by showing very small random changes in HR, which is our main outcome, in response to 4X4. Thank you and we hope that this explanation appeases your concern.

Results: Figure 1 shows that the HR response to each interval. Your point is well-taken as these values look similar, when in fact they are not. We agree with your assertion, so text was added here to further clarify the HR response to each interval during 4X4; we hope that this addition assuages your concern; thank you. Figure 1 shows participants’ average HR response to each interval with mean HR equal to 160.3 ± 11.0, 166.5 ± 11.0, 167.0 ± 10.0, and 169.0 ± 10.0 b/min, respectively, during each interval.

Figure 1: Also note that we added brief language to be more clear in the Figure legend. As to why our text stated that HR increased during the 4X4 protocol, this is standard language reflecting a significant main effect of time which we believe needs to be denoted here. Thank you and we hope that this satisfies your concern.

Time spent between 85 – 95 %HRmax: Your point here is fair, yet we believe that text is more than adequate to report this little bit of data, which we do not believe merits its own table.  We hope that this explanation appeases your concern; thank you.

We also believe that Figure 1a is adequate the way it is, and Figure 2a too, as both show the upward trend in HR and decrement in PO during this protocol. These are two of the most important results that we want to emphasize in this paper, so we believe these data are fine as presented. 

Figure 1b: We have revised this Figure per your comments and we hope that it is now satisfactory; thank you.

All other Figures have been slightly revised per your comments; thank you.

Discussion:  Your point is well-taken, but we still do believe that one limitation of the majority of studies using the 4X4 protocol is that insufficient text is presented in the Methods to allow replication, which goes against what this section is supposed to achieve.  Yet, we did change some of our text to reflect this, and we hope that this satisfies your concern.

Your point to do additional testing to attempt to identify an ‘optimal’ 4X4 protocol is appreciated, yet we lacked the time and resources to allocate this much effort to this particular study. We hope this explanation appeases your concern; thank you.

Reviewer 3 Report

The manuscript is focused on a interesting topic and, although the results are not exactly innovative, the main findings are sounding. I have just some considerations the Authors should address:

1) is the selected sample size based on an "a priori" analysis? If so, please specify.

2) why did the Authors choose a ramp protocol instead of an incremental square wave test?

3) Please provide more details about the activity performed by the participants.

3) Why was ventilatory threshold calculated if it is not considered in Discussion?

4) Your data presents an interesting description of the characteristics (under an HR work point of view) in acute. But, how much is translatable during repeated measures during training?

Author Response

Reviewer #3, Specific comments: We thank this reviewer for his/her positive comments regarding our work.

  1. Sample size: We did not do any sample size or power analysis as we were confident that our sample size is sufficient to detect small changes in our measures. This sample is two times larger than other papers in this area (References 14 and 18), and in addition, we performed reliability testing to show that the day-to-day variability in HR was really low (< 2 b/min). So, we are confident that we had adequate statistical power to detect small differences in our outcomes. Thank you and we hope that this explanation appeases your concern.
  2. VO2max protocol: The Corresponding Author has used this particular incremental protocol characterized by ramp-like changes in work rate each minute (e.g. 25 W/min = 1 W every 2.4 s) for almost 15 yr to assess VO2max and PPO, and moreover, to measure changes in these parameters in response to chronic training. VO2max is extremely reliable across days using this protocol (CV ~ 3 %) and it is well-tolerated and time-efficient (8 – 12 min). So, we really see no need to utilize any other exercise testing protocol than this one. We have used square wave intensities above that associated with VO2max to confirm incidence of VO2max (reference 7), but this procedure is relegated to studies where we monitor changes in VO2max in response to exercise training. We appreciate your point and we hope that this clarification satisfies your concern; thank you.
  3. This is a fair point you raise here. All men and women performed > 150 min/wk of physical activity including resistance training, aerobic exercise, non-competitive sport, and rowing, although none was athletic or training for a particular sport. Text has been added here per your helpful comment; thank you. These adults participated in resistance training, aerobic exercise, non-competitive sport, and/or rowing, yet none was athletic or training for a particular sport.
  4. This is a great point. We initially believed that time spent above 85 %HRmax would be associated with VT, which is related to exercise tolerance as it is represented as the transition from purely oxidative to a greater contribution of non-oxidative metabolism to ATP supply during exercise, which is attendant with HIIT. However, there was no association between these variables, which was surprising. Due to this non-significant result, we did not believe that this text merited additional explanation in the Discussion. Overall, we hope that you approve of this explanation; thank you.
  5. You raise a fair point here. We would expect that chronic use of 4X4 as part of a longitudinal exercise intervention would require gradual increases in PO during daily sessions to keep participants in the target HR zone, considering the reduction in submaximal HR typically recorded in response to training as well as the well-known increase in PPO (reference 7). We could add this somewhere in the Discussion, yet it may be too speculative and will leave it as is; we hope that this is satisfactory.

Round 2

Reviewer 2 Report

Dear Authors,

I have read your rebuttal with great surprise. I don't know if you are familiar with responding to reviewers, but, as far as I am concerned, you should address each issue and either give enough rationale against or change some bits on the manuscript, especially when there is a major changes petition.

Your response follows the  following structure: 

Yes, we agree with the reviewer in bla bla, yet we are not changing anything,  we hope that this explanation appeases your concern.

Do you really think that by giving a few words with no enough solid foundation you will be convincing a reviewer who has spent a lot of time examining your work? Out of 16 points raised by the reviewer, you only changed 5 bits of the manuscript with merely cosmetic changes (remember it was a major changes revision). And the worst part is that your are not right in most of your arguments. Take the first point:

"Abstract. Remove subtitles (objectives, methods…)"

and your answer:

"Our understanding is that the Journal Guidelines require these subheadings here, so we do not believe it is proper to remove them. Thank you."

Have you read the Instructions for Authors, where it is stated: "The abstract should be a single paragraph and should follow the style of structured abstracts, but without headings: 1) Background: Place the question..."? 

I strongly suggest you pay more attention to any reviewer's concerns and try to convince him/her of the quality of your work but also to accomplished changes asked and also by elaborating more on your response.

Unfortunately, you leave me no other option but to reject this manuscript without further revision.

Author Response

Additional comments to Reviewer #2:  It is unfortunate that Reviewer #2 still has remaining concerns regarding our submission. Below, we have attempted to respond to his/her concerns and newly-written text is placed in the manuscript in red font. And please note that the Authors do reserve the right to not make changes to a paper based on referees’ comments when we believe that our position is right and substantiated by best practices in the field.  And note that the Corresponding Author has published approximately 80 peer-reviewed manuscripts, so he is well-versed in how to revise a manuscript and construct a point-by-point rebuttal.  We do not believe that the format of our prior rebuttal is condescending or dismissive to this particular reviewer; in fact, we value all feedback regarding our work as overall, it is beneficial to strengthening the quality of the submission.   

Format of Abstract:  We made an error here as we consulted a recent manuscript published in the Journal that contained subheadings in this section. We have consulted the Instructions for Authors, and it does say that subheadings are not required, as you cite, so they were removed from the text.  We hope that this correction appeases your concern.

Sufficient evidence of HIIT’s benefits vs. MICT: We reported data from 2 review articles here to be concise and not present too much background text, as there are more than 10 yr of data showing the efficacy of HIIT compared to MICT in many diverse populations.  We believe that anyone who reads this paper is well-aware of this vast data supporting the efficacy of HIIT on various health-related outcomes.  That said, per your comment, we added text to further emphasize the superior benefits of HIIT vs. MICT, and we hope that this addition appeases your concern.

Use of a standard HIIT protocol: So we read your original comments here, and it is very hard to understand what you are trying to convey here, and if you want us to make any changes to the text or merely explain our narrative in this section, which is what we did in our initial rebuttal.  The point of this manuscript is that despite 4X4 being widely used in research and clinical settings, its ease of implementation is less than that of other protocols based on PPO (the 10 X 1 min regimen) or some fraction of body mass (REHIT and repeated Wingate tests). Moreover, most studies do not fully explain in their Results how PO is modified during 4X4 to attain the target HR, which is a disservice to the audience as it does not allow replication.  We did revise and add some additional text here to strengthen our narrative, and we hope that this appeases your concern; thank you. 

And yes, you are correct that our results only apply to the participants tested and methods employed, and we agree that it does not solve the problem of not having a standardized 4X4 regimen. However, part of why HIIT has become such a popular topic in sports science is because of the marked increase in studies investigating its effects, which then allows others to employ previously-utilized protocols in their own settings. We hope that this explanation appeases your concern; thank you.

Table 1: Please note that these are not athletes but habitually active men and women. That said, we are comparing their physical traits and showing data for both men and women as this is typical practice when using mixed gender samples in a study like this one.  Our Results do show that women spent more time in the desired HR zone than men, which suggests presence of different responses to 4 X 4 based on sex.  The other Reviewers had no issue with this presentation of data, so we believe it is proper as is.

Three men doing reliability testing: Only these 3 men whose age, body mass, and VO2max were similar to the entire sample performed these 3 days of testing. We believe to best characterize the HR and PO response to 4 X 4 in an attempt to develop a protocol that can be used in a similar population, we need to have some knowledge of the day-to-day variability in these measures, which is what these Results give us. Brief text was added to line 138 to better explain this; thank you.

Figure 1: To better clarify these results, mean ± SD data were added to the initial revision per your comment.  Also, new text was added to the paper to clarify that post hoc analyses showed that HR in intervals 2, 3, and 4 was higher than for interval 1, and keep in mind that the Cohen’s d value here reflects that this is a ‘large’ effect or difference.  We hope this addition appeases your concern. And the reference to p < 0.001 denotes the main effect of time on HR during the session. Yes, it is anticipated that HR will significantly rise during any acute HIIT session; however, we wanted to be candid to the reader and we believe this text achieves this.

Figure 2: We added the mean ± SD value for power output in the text here (lines 167-168), per your comment. We believe that this is a suitable alternative to a table and accomplishes what it seems you desire, which is to show actual data values here.

And your comment below “Finally, consider the 85 – 95 %HRmax individually against the individual response and depict mean” is unclear to us, so we did not make any modifications to the manuscript in this area.

Discussion: Text was added in lines 190-191 to better articulate limitations of prior work that merits studies like the current one.

Overall, we read the Discussion twice per your comment, and tried to ensure that the text did not suggest that our study is a solution, but instead, simply disseminates data that add to what is known regarding constructing the 4 X 4 protocol. We hope this is satisfactory.

Round 3

Reviewer 2 Report

Additional comments to Reviewer #2:  It is unfortunate that Reviewer #2 still has remaining concerns regarding our submission. Below, we have attempted to respond to his/her concerns and newly-written text is placed in the manuscript in red font. And please note that the Authors do reserve the right to not make changes to a paper based on referees’ comments when we believe that our position is right and substantiated by best practices in the field.  And note that the Corresponding Author has published approximately 80 peer-reviewed manuscripts, so he is well-versed in how to revise a manuscript and construct a point-by-point rebuttal.  We do not believe that the format of our prior rebuttal is condescending or dismissive to this particular reviewer; in fact, we value all feedback regarding our work as overall, it is beneficial to strengthening the quality of the submission.   

Dear Authors,

You are right that you can reserve the right not to make the changes proposed by the reviewer, just as the reviewer has the right to reject the lack of changes when they are not substantiated. And when it comes to the number of peer-reviewed manuscripts, this reviewer has a similar number with over 20 years of research experience, so I don't think the authors are in a position to teach any lessons. Finally, your rebuttal has been dismissive indeed to me, and just because you value my suggestions doesn't necessarily mean I have to admit that you reject them.

That said, I am glad you have resumed the revision where it was left in the first round and I appreciate the changes made, so I will conduct an honest revision of the manuscript in its current state.

Format of Abstract: I am pleased to see that you have finally removed the subheadings.

Sufficient evidence of HIIT’s benefits vs. MICT: I am pleased to see that you have indicated two studies that show the efficacy of HIIT compared to MICT in many diverse populations.

In my first round of review, I asked what was the point of comparing male and female athletes? Your answer indicates that you want to know if there are differences between men and women, given that they are not professional athletes. This fact does not justify applying a statistical test (which on the other hand we do not know) to find out if there are statistically significant differences between the sexes. My suggestion is to remove this statistical comparison and leave the descriptive data for both groups. Finally, I do not know what the opinion of the other reviewers has been, nor the value that you assign to their comments compared to mine, but I reiterate the request expressed in this paragraph and kindly request the described change.

Regarding the three men doing reliability testing issue, please include the arguments put forth in your rebuttal in the manuscript text, as it clarifies to the reader the reason for using only three men.

Figure 1. In my first round of reviews, I asked to re-elaborate Figure 1b with more accuracy on the HR line (thick line with little detail) and better axis. You have not answered anything about this aspect, so I consider that you agree. However, there is no change in Figure 1b. Please make the change.

Author Response

The Authors have responded to your final concerns in the point-by-point rebuttal below, which we hope is satisfactory. Changes have also been made to the manuscript in red font as well.

  1. Table 1 was changed and no longer has the data separated by sex, per your comment.
  2. Text was added to better describe the 3 men who performed reliability testing. As for why only 3, we really do not believe we need to explain why this number compared to 4 or 5 was chosen. Most studies like this one, as we cite, present no reliability data at all for the change in HR or PO, so we feel that this set of data informs the reader as to the random change in these variables in response to 4 X 4 HIIT.
  3. We slightly revised the y-axis of Figure 1b although it is still unclear what is meant by 'better axis' and 'more accuracy' in your comments below. The other Reviewers did not have any concerns with its format, which we believe follows norms for Figures such as this presenting HR data. We hope that this is satisfactory.